# The Relationship Between Emotional Intelligence and the Risk of Eating Disorders Among Adolescents: The Mediating Role of Motivation for the Use of Social Media and Moderation of Perceived Social Support

**DOI:** 10.3390/bs15040434

**Published:** 2025-03-28

**Authors:** Martina Riolo, Marco Andrea Piombo, Vittoria Spicuzza, Cinzia Novara, Sabina La Grutta, Maria Stella Epifanio

**Affiliations:** Department of Psychology, Educational Science and Human Movement, University of Palermo, 90128 Palermo, Italy; martina.riolo@unipa.it (M.R.); vittoria.spicuzza@unipa.it (V.S.); cinzia.novara@unipa.it (C.N.); sabina.lagrutta@unipa.it (S.L.G.); mariastella.epifanio@unipa.it (M.S.E.)

**Keywords:** trait emotional intelligence, eating disorders, adolescents, social media motivation, perceived social support, mediation, moderation

## Abstract

Trait Emotional Intelligence (Trait EI) is considered a protective factor for adolescents’ psychological well-being and may play a critical role in mitigating the risk of developing eating disorders (EDs), particularly in the context of pervasive social media use. However, the psychological mechanisms underlying this relationship, such as the driving factors of social media engagement, remain underexplored. This cross-sectional study aimed to examine whether motivating factors for social media use mediate the relationship between Trait EI and ED risk, as well as whether perceived social support moderates this relationship. A total of 388 Italian adolescents (Mage = 14.2; 50.7% girls) completed self-report questionnaires, including the Trait Emotional Intelligence Questionnaire—Adolescent Short Form (TEIQue-ASF), the Eating Attitudes Test (EAT-26), the Multidimensional Scale of Perceived Social Support (MSPSS), and the Motivations for Social Media Use Scale (MSMU). Data were collected between November 2023 and June 2024. Mediation and moderation analyses were conducted using the PROCESS macro (Model 5). The results showed that lower Trait EI scores were significantly associated with higher EAT-26 scores (β = −11.03, *p* < 0.001). Motivation for social media use in terms of popularity, (β = −0.35, *p* < 0.05), appearance (β = −0.68, *p* < 0.01), and connection (β = −0.44, *p* < 0.05) significantly mediated this relationship. Perceived social support moderated this relationship in all models (β range = 0.08–0.10, *p* < 0.05), suggesting a buffering effect. The findings highlight the importance of Emotional Intelligence and social support as key psychological resources that may protect adolescents from disordered eating behaviors. Moreover, understanding the motivating factors behind social media use, particularly those centered on appearance and popularity, may help identify adolescents at greater risk and inform tailored prevention strategies.

## 1. Introduction

Emotional Intelligence (EI) is a broad construct that refers to the ability to perceive, understand, regulate, and manage emotions in oneself and others. Within the EI framework, two primary conceptualizations have emerged: Trait EI and Ability EI ([29]; [25]). Trait EI is defined as a set of emotion-related self-perceptions and dispositions, which are typically assessed through self-report measures and linked to personality traits ([31]). Differently from Ability EI, which focuses on cognitive-affective processing and is measured through performance-based assessments, Trait EI captures stable emotional characteristics that shape how individuals experience and regulate their emotions in daily life ([39]).

A growing body of research suggests that Trait EI plays a crucial role in psychological well-being, acting as a protective factor against mental health disorders and maladaptive behaviors in adulthood ([38]; [24]) and in adolescence ([17]). More specifically, individuals with high Trait EI tend to employ adaptive emotional regulation strategies, which help them maintain positive emotional states and mitigate the impact of stressors ([2]; [5]; [12]). Conversely, low Trait EI has been associated with higher vulnerability to mood disturbances, anxiety, and maladaptive coping mechanisms, including dysfunctional eating behaviors both in adults and adolescents ([6]; [10]). Regarding this, given the well-established link between emotion regulation deficits and eating disorders (EDs) ([1]), two recent systematic reviews further confirm this relationship, highlighting the negative association between Emotional Intelligence (EI) and various dimensions of EDs, emphasizing mechanisms such as adaptability, stress tolerance, and emotional regulation ([36]; [45]), and in this regard, Table 1 summarizes the differences between Ability EI and Trait EI, along with their relationships with EDs.

Going more in-depth, empirical evidence suggests that individuals with low EI may struggle with identifying and regulating negative emotions, which could increase the likelihood of engaging in disordered eating as a maladaptive coping strategy ([15]). Specifically, binge-eating episodes and restrictive eating behaviors have been linked to difficulties in processing emotions and reduced emotional self-efficacy ([36]). Moreover, deficits in emotional processing can heighten susceptibility to external influences, including media-driven body image ideals and social comparison processes ([19]).

In recent years, the exponential growth in social media usage has raised concerns about its potential impact on adolescents’ mental health and eating behaviors. With approximately 60% of the global population actively engaged on social media platforms ([23]), adolescents represent a particularly vulnerable population due to their ongoing identity formation and heightened sensitivity to social validation ([44]; [32]). Research indicates that prolonged exposure to idealized body images on social media is associated with increased body dissatisfaction, self-esteem issues, and engagement in disordered eating behaviors, particularly among young females ([13]; [27]).

However, while much of the existing literature focuses on the negative consequences of social media, recent studies highlight the importance of understanding the motivations behind adolescents’ engagement with these platforms. Rather than the mere duration of use, the underlying psychological factors driving social media consumption—such as the need for social approval and emotion regulation—may play a more significant role in shaping adolescents’ experiences and outcomes ([41]). In this context, Trait EI may influence how adolescents interact with social media and whether their engagement fosters adaptive or maladaptive psychological outcomes ([43]).

Another key factor influencing adolescents’ emotional functioning is perceived social support (PSS), which has been consistently linked to improved well-being and lower risks of internalizing disorders such as depression and anxiety ([8]; [37]). During adolescence, peer support and validation become particularly salient, and social media platforms serve as a primary medium for seeking feedback and connection ([9]). While high PSS has been associated with greater emotional stability and life satisfaction ([22]), the absence of meaningful peer interactions online may exacerbate emotional distress and reinforce maladaptive behaviors, including disordered eating ([4]).

While prior research has examined the relationships between social media use, EDs, and Trait EI, there remains a significant gap in understanding the role of motivating factors for social media use and the nature of the content consumed. The current literature has primarily focused on quantitative aspects of social media engagement, such as screen time, rather than investigating the psychological drivers behind social media use and how they mediate the relationship between EI and ED risk.

### The Present Study

This cross-sectional study aims to examine the role of social media use motivation and perceived social support in the relationship between Trait EI and the risk of developing EDs. Specifically, we hypothesize that motivating factors for social media use act as mediators in this relationship. Individuals with low Trait EI may engage with social media for maladaptive reasons, such as seeking external validation or managing negative emotions, which in turn could increase their vulnerability to disordered eating behaviors. Additionally, we hypothesize that perceived social support acts as a moderator in this relationship (Figure 1). High levels of perceived social support may buffer against the negative effects of maladaptive social media use, thereby reducing the impact of low Trait EI on eating disorder risk. Conversely, low perceived social support may exacerbate emotional vulnerabilities, reinforcing the cycle of social media-driven body dissatisfaction and disordered eating behaviors. By addressing these gaps, the present study seeks to advance our understanding of the theoretical relationships between Trait EI, social media motivations, and perceived social support. Ultimately, our findings aim to inform the development of targeted interventions that can reduce the risk of disordered eating behaviors in adolescents.

## 2. Materials and Methods

### 2.1. Participants and Procedures

The participants were adolescents from twelve classes, each from two Italian public secondary schools, both located in urban areas. The inclusion criteria required participants to be between the ages of 13 and 17 years and to have no prior diagnosis of eating disorders. The exclusion criteria included students with a history of eating disorders or those who did not provide parental consent. Permission for the study was first obtained from the school principals, followed by active signed consent from parents. The final sample comprised 388 adolescents, balanced between boys and girls (50.7% girls), with an average age of 14.2 years (SD = 0.72; age range = 13–15 years). An a priori power analysis conducted using G*Power 3.1 indicated that a sample size of approximately 200 participants would be sufficient to detect a moderate effect size (f^2^ = 0.05) in a multiple regression model with three predictors at an alpha level of 0.05 and 80% statistical power. Given our sample size of 388 participants, the study had adequate power to detect small-to-moderate effects. Data collection took place between November 2023 and June 2024. Participants completed online anonymous self-report questionnaires during a regular school day in the presence of their teacher. Before data collection, they were reassured about confidentiality and informed that they could withdraw from the study at any time without consequences. The survey took approximately 30 min to complete. After completing the questionnaires, participants were thanked for their participation, and researchers addressed any questions raised. While public involvement was not directly included in the study design, school administrators and teachers were consulted throughout the research process, particularly in the recruitment phase and the facilitation of data collection within the schools.

All procedures complied with the ethical standards of the relevant national and institutional committees on human experimentation and with the Helsinki Declaration of 1975 (revised in 2008). The study was approved by the Bioethics Committee of the institution of the first author (Prot. n.116/2022).

### 2.2. Measures

#### 2.2.1. Risk of Eating Disorders

The Eating Attitude Test-26 (EAT-26; [16]; [11]) in its Italian version was used to assess eating disorder symptoms. It consists of 26 items, scored on a 6-point Likert scale ranging from always to never. The questionnaire measures three primary dimensions: dieting, bulimia and food preoccupation, and oral control. The total score ranges from 0 to 78, and a score equal to or higher than 20 indicates that a subject may be at risk of an eating disorder. Higher scores reflect more severe symptomatology. The EAT-26 has been widely applied to adolescent populations to identify early signs of disordered eating behaviors. Recent research confirms its reliability and validity in different cultural settings, and the reliability of this study was good: Cronbach’s alpha = 0.85.

#### 2.2.2. Perceived Social Support

The Multidimensional Scale of Perceived Social Support (MSPSS; [46]) was used to assess perceived social support. This is a 12-item scale that measures perceived social support from three sources: Family, friends, and significant others. The scale was rated in a seven-point Likert response format (1  =  “very strongly disagree” to 7  =  “very strongly agree”). The total scores range from 12 to 84, with higher scores indicating greater total perceived social support from all three sources. The internal reliability of the scale in our sample was good: Cronbach’ alpha = 0.89.

#### 2.2.3. Trait Emotional Intelligence

The Trait Emotional Intelligence Questionnaire—Adolescent Short Form (TEIQue-ASF; [30]) was used for this study. We used the Italian version of the TEIQue for adolescents as the basis for our measure. Specifically, we derived the short-form version (TEIQue-ASF) by utilizing the Italian translation of the original short form. The translation was carefully reviewed and adapted for this study to ensure conceptual equivalence with the validated full version for Italian adolescents.

The Italian adaptation of the TEIQue for adolescents has been previously validated, demonstrating good psychometric properties and internal consistency ([3]). This validation confirmed the appropriateness of the TEIQue framework for assessing Trait EI in Italian adolescents, supporting its relevance in research on emotional regulation, social interactions, and psychological well-being. The ASF comprises 30 short statements on a 7-point Likert scale designed to measure global Trait EI and the four broad factors of Trait EI: well-being, self-control, emotionality, and sociability. For the purpose of this study, only global Trait EI scores were used, and the questionnaire showed good internal reliability in our sample: Cronbach’s alpha = 0.86.

#### 2.2.4. Motivations for Social Media Use

The Motivations for Social Media Use Scale (MSMU; [34]) was used to assess motivations for social media use. This is a self-report questionnaire on a Likert scale designed to assess the psychological and behavioral motivations that drive individuals to engage with social media platforms ([34]). The scale identifies key motives underlying social media use, including social interaction, self-expression, entertainment, and information-seeking. The MSMU consists of multiple subscales, each measuring a distinct motivation for social media use, including “connection” (when social media is used to maintain relationships and interact with others), “popularity” (when social media is used for seeking social validation and online recognition), appearance (engaging with social media to compare and enhance physical self-presentation), and values and interests (using social media to explore personal interests and identity development). Since no validated Italian version of the MSMU currently exists, for this study, we back-translated the original English version into Italian. The reliability for this study ranges from acceptable to good for each subscale (connection = 0.63; popularity = 0.82; appearance = 0.84; values and interests = 0.71).

### 2.3. Statistical Analysis

All statistical analyses were conducted using SPSS version 25. First, preliminary analyses, including Pearson’s correlation (r), were conducted to examine the relationships between Trait EI, motivation for the use of social media, perceived social support, and the risk of developing eating disorders. Additionally, multiple two-way ANOVAs were conducted to assess potential gender differences across these variables. Secondly, to examine the proposed mediation and moderation effects, we employed Hayes’ PROCESS macro for SPSS ([20]). Specifically, we tested a moderated mediation model (Model 5) in which Trait EI was input as the predictor variable, the risk of developing eating disorders as the outcome variable, and motivation for social media use as the mediator. Perceived social support was included as a moderator of the relationship between motivation for social media use and eating disorder risk. We conducted separate mediation and moderation analyses for each subscale of social media motivation to examine their distinct contributions to the model. Bootstrapping with 5000 resamples was utilized to generate confidence intervals for indirect effects, ensuring robustness in estimating mediation and moderation effects. The significance of effects was determined using 95% confidence intervals, where intervals not crossing zero indicated significant mediation or moderation effects.

## 3. Results

### 3.1. Descriptive Statistics and Sex Differences

Descriptive statistics and sex differences for the study variables are reported in Table 2.

Specifically, regarding screen time, the majority of adolescents (46.0%) reported spending between 3 and 5 h per day on social media. Additionally, 25.0% of participants spent 6 to 8 h daily. A smaller proportion, 10%, reported using social media for 9 to 11 h per day, while 4.9% of adolescents exceeded 12 h per day. In this regard, a significant sex difference was found, F = 24.21, *p* < 0.001, with males reporting lower screen time compared to females. For Trait EI, boys scored significantly higher than girls: F = 17.26, *p* < 0.001. In terms of perceived social support (PSS), both boys and girls reported comparable levels with no significant sex differences, F = 1.32, *p* > 0.05. Significant sex differences were observed in eating disorder risk, χ^2^ = 31.01, *p* < 0.001. A greater proportion of girls (32.3%) exceeded the clinical cut-off for eating disorder risk, compared to only 9.3% of males. Furthermore, among adolescents classified as at risk for EDs, the majority (47.6%) reported spending between 3 and 5 h online daily, while 25% reported 6 to 8 h, and 14.2% exceeded 12 h per day. Similarly, those who were not at risk predominantly reported mainly between 3 and 5 h per day (45.5%), with only 9.9% exceeding 12 h per day. With regard to motivation for social media use, girls reported significantly higher scores than males in the connection (F = 13.77, *p* < 0.001), popularity (F = 8.27, *p* < 0.01), and appearance (F = 12.77, *p* < 0.001) subscales, while no significant gender differences were found in the values and interest subscales (F = 0.42, *p* > 0.05).

### 3.2. Correlational Analysis

Bivariate associations between variables are reported in Table 3. Specifically, screen time showed a statistically significant positive correlation with eating disorder risk scores (r = 0.15, *p* = 0.003) and with some motivating factors for social media use, particularly connection (r = 0.21, *p* < 0.001), popularity (r = 0.24, *p* < 0.001), and appearance (r = 0.19, *p* < 0.001), while a significant negative association was found with Trait EI (r = −0.26, *p* < 0.001) and no significant association was found with perceived social support (r = −0.04, *p* > 0.05).

Trait EI global scores were negatively correlated with eating disorder risk (r = −0.38, *p* < 0.001) and with motivations for social media use, including connection (r = −0.29, *p* < 0.001), popularity (r = −0.18, *p* < 0.001), and appearance (r = −0.31, *p* < 0.001), while a positive association was found with perceived social support (r = 0.48, *p* < 0.001). Eating disorder risk scores were positively correlated with motivating factors for social media use, particularly appearance-based motivation (r = 0.27, *p* < 0.001), popularity motivation (r = 0.21, *p* < 0.001), and connection motivation (r = 0.22, *p* < 0.001), and negatively correlated with perceived social support (r = 0.15, *p* < 0.01) Finally, perceived social support was only negatively associated with appearance (r = −0.15, *p* = 0.003) among motivating factors for social media usage, while no association was found with other motivating factors (*p* > 0.05).

### 3.3. Mediation and Moderation Effects

With regard to the mediation effects of different motivating factors for social media use and the moderation effect of social support, for the relationship between Trait EI and ED risk, the results of each model are presented in Table 4. Specifically, in Model 1, the direct effect of Trait EI on eating disorder risk was significant (β = −10.92, SE = 3.05, *p* < 0.001), while the mediation analysis revealed that popularity motivation for social media use partially mediated the relationship (β = −0.35, SE = 0.15, *p* < 0.05) and perceived social support significantly moderated this relationship (β = 0.08, SE = 0.04, *p* < 0.05). In Model 2, which included appearance-based social media motivation, the direct effect of Trait EI on eating disorder risk remained significant (β = −11.03, SE = 3.05, *p* < 0.001). The mediation path through appearance motivation was significant (β = −0.68, SE = 0.22, *p* < 0.01), while perceived social support showed a small but significant moderation effect (β = 0.09, SE = 0.04, *p* < 0.05). In Model 3, the same pattern was observed for connection motivation, where Trait EI negatively predicted eating disorder risk (β = −11.22, SE = 3.08, *p* < 0.001), and the mediation pathway through connection motivation was significant (β = −0.44, SE = 0.20, *p* < 0.05). Also, in this model, the moderation effect of perceived social support was significant (β = 0.09, SE = 0.04, *p* < 0.05). In Model 4, which tested the motivation of interest/values for social media use as a mediator, the mediation path was not significant (β = 0.01, SE = 0.10, *p* > 0.05). However, the direct effect of Trait EI remained significant (β = −12.51, SE = 3.05, *p* < 0.001), while the moderation effect of perceived social support was also significant (β = 0.10, SE = 0.04, *p* < 0.05). Across all models, the total indirect effect of Trait EI on eating disorder risk through social media motivation and perceived social support was significant in the popularity, appearance, and connection models (*p* < 0.05) but not in the interest/values model (*p* > 0.05). The proportion of variance explained by the models (R^2^ values) ranged from 17% to 19%, indicating a moderate effect size.

## 4. Discussion

The present study explored the relationships among Trait EI motivations for social media use, perceived social support, and adolescent ED risk. The preliminary findings largely align with the previous literature, particularly regarding sex differences in Trait Emotional Intelligence and eating disorder risk. However, certain results offer novel perspectives into the mediating role of social media motivation and the moderating effect of perceived social support. First of all, females reported higher levels of ED risk than males, supporting previous findings that showed that being an adolescent girl is associated with have more disordered eating behaviors, and this association could be explained by sociocultural pressures, body dissatisfaction, and engagement in appearance-driven social media use ([13]; [18]). With regard to sex differences in Trait EI, our results showed that males scored higher in global Trait EI compared to females, which is in line with some prior research suggesting that boys may be better at regulating their own emotions, while girls could be better at recognizing and managing the emotions of others ([33]). However, other studies challenge this perspective, indicating that gender differences in Trait EI are not always significant and may be context-dependent ([14]). This inconsistency in the literature suggests that the relationship between sex and Emotional Intelligence is influenced by multiple factors, including socialization processes and the different kinds of measures used ([40]).

Our findings also highlight the significant association between screen time and eating disorder risk. This result is consistent with previous research linking excessive social media use with increased body dissatisfaction, social comparison, and engagement in maladaptive eating behaviors ([28]). Moreover, our results showed a negative association between Trait EI and screen time, supporting the hypothesis that adolescents with lower Emotional Intelligence may use social media for longer, maybe as an external regulatory strategy for managing emotions ([7]). Individuals who struggle with emotional regulation or have difficulty mentalizing their emotions may be more inclined to engage in prolonged social media use as a form of emotional escape or a self-regulation mechanism using external devices.

The mediation and moderation models suggested, in general, that it is not just the amount of time spent on social media but rather the motivating factors underlying social media use that play a crucial role in explaining the relationship between Emotional Intelligence and eating disorder vulnerability. Specifically, Trait EI is negatively associated with specific motivations for social media use, particularly popularity and appearance, which in turn increase the risk of developing eating disorder symptoms. From this perspective, it is not surprising that motivations linked to physical appearance and the desire for social approval turned out to be factors mediating the relationship between Trait EI and the risk of eating disorders, whereas other motivations (such as values and interests) do not show a significant impact. This is in line with the notion that adolescents who engage with social media for self-presentation and social validation could be at a higher risk of disordered eating behaviors ([35]). Moreover, this aligns with prior studies suggesting that appearance-focused social media engagement fosters body dissatisfaction and maladaptive eating patterns ([42]). The fact that popularity motivation also mediated this relationship could indicate that social comparison processes and peer validation pressures may further exacerbate disordered eating tendencies ([35]). Aiming to go more in-depth into these concepts, we can hypothesize that lower Trait EI predisposes adolescents to a reliance on social media as an external regulator of self-esteem, seeking likes, comments, or other forms of validation to compensate for difficulties in managing emotions and devaluated self-perception. Such reliance on external cues can exacerbate body dissatisfaction and self-comparison, both recognized as risk factors for disordered eating ([35]). Conversely, adolescents with higher Trait EI tend to regulate their emotions more autonomously and probably maintain a more stable sense of self-worth, thereby reducing their need for constant online approval. Moreover, the moderating role of perceived social support suggests that adolescents who feel supported by family, friends, or significant others may be less inclined to seek self-esteem boosts through social media, thus further protecting them from the risk of eating disorders. Specifically, higher levels of social support appeared to buffer the negative impact of low Emotional Intelligence, possibly promoting greater emotional resilience and reducing the need to seek validation through social media. Adolescents with strong perceived social support networks may be less likely to isolate themselves from real-world interactions, decreasing their reliance on social media as an external source of self-esteem enhancement, showing less dependence on virtual feedback, and benefiting from a “protective buffer” against the risk of internalizing unrealistic and harmful models proposed by social networks ([21]).

In conclusion, these results may have many practical applications. Promoting Trait EI in adolescents (for instance, through emotional literacy training and programs aimed at enhancing socio-emotional skills) could reduce the need to seek external validation via social media constantly. Such an intervention, combined with the promotion of more awareness about the use of digital platforms, may help prevent the vicious cycle of social comparison and low self-esteem, reducing the risk of developing eating disorders. However, given the sample size limitations and the observed variability in effect estimates, these findings should be interpreted with caution and considered as preliminary evidence that requires further validation in larger-scale studies.

### Limitations and Future Directions

Although this research contributes to the advancement of the relationships between Trait EI, social media motivations, perceived social support, and eating disorder risk, several limitations should be noted. First, the cross-sectional design does not allow for definitive causal inferences regarding the observed associations, because different confounding variables could not be accounted for. Second, despite the fact that our sample size (N = 388) was adequate for detecting moderate effects, the variability in confidence intervals suggests that a larger sample could improve the precision and stability of the findings. Third, the use of only self-report measures raises the possibility of response bias, including social desirability and inaccuracies in recalling social media usage. Fourth, the sample was drawn from a specific cultural and geographical context, potentially limiting the generalizability of the findings to other populations. Fifth, although screen time was considered, the study did not capture the more qualitative aspects of social media use (e.g., specific platforms, content types, or interaction patterns) that could further elucidate the mechanisms linking social media motivations to disordered eating. Finally, additional factors such as personality traits or peer influences, which may also shape adolescents’ online behaviors and vulnerability to eating disorders, were not included in the present analysis. Future research would benefit from employing longitudinal designs, incorporating more diverse samples, and examining a broader range of variables to deepen our understanding of these complex interrelationships.

## Figures and Tables

**Figure 1 behavsci-15-00434-f001:**
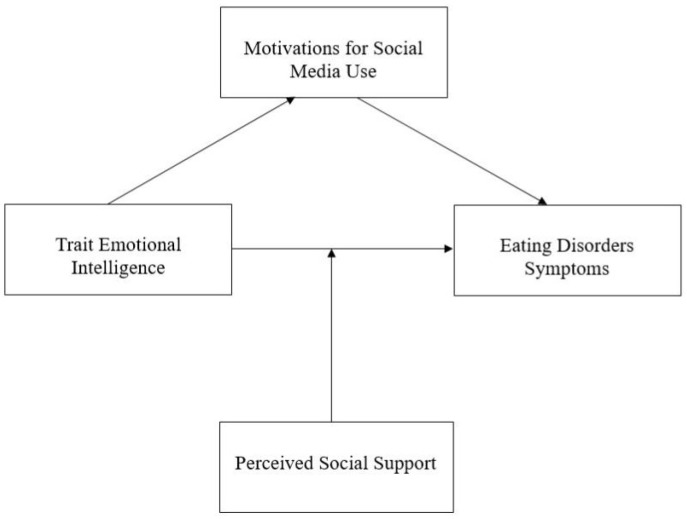
The hypothesized mediation–moderation model.

**Table 1 behavsci-15-00434-t001:** Overview of Trait and Ability EI in relation to EDs.

	Trait EI	Ability Emotional Intelligence (Ability EI)
Definition	Self-perceived emotional abilities and dispositions ([29])	Cognitive–emotional abilities measured through objective tasks ([26])
Measurement	Self-report questionnaires (e.g., TEIQue)	Performance-based tests (e.g., MSCEIT)
Association with EDs	Stronger negative correlation with disordered eating behaviors (r = −0.23, *p* < 0.001)	Weaker negative correlation with disordered eating (r = −0.12, *p* = 0.010)
Mechanisms linking to ED risk	Emotion dysregulation, stress vulnerability, reliance on external validation ([45])	Limited influence on ED risk, as it mainly involves emotion-related cognitive processes ([45])

**Table 2 behavsci-15-00434-t002:** Descriptive information and sex differences in the sample.

	Females (*n* = 195)	Males (*n* = 193)	Total (*n* = 388)	F/χ^2^
Screen Time ^a^				24.21 ***
0–2 h/day	17 (8.71)	38 (19.68)	55 (14.1)	
3–5 h/day	81 (41.53)	97 (50.25)	178 (46.0)	
6–8 h/day	54 (27.70)	43 (22.28)	97 (25.0)	
9–11 h/day	29 (14.87)	10 (5.18)	39 (10.0)	
>12 h/day	14 (7.18)	5 (2.6)	19 (4.9)	
Trait EI ^b^				17.26 ***
Global Score	4.51 (±0.77)	4.83 (±0.76)	4.67 (±0.88)	
Perceived Social Support ^b^				
MSPSS global score	68.10 (±13.16)	66.51 (±13.98)	67.31 (±0.88)	1.32
EDs Risk ^a^				31.01 ***
<Cut-Off	132 (67.70)	175 (90.67)	307 (79.1)	
>Cut-Off	63 (32.32)	18 (9.32)	81 (20.9)	
Motivation for Social Network ^b^				
Connection	8.07 (±2.94)	7.00 (±2.73)	7.54 (±2.89)	13.77 ***
Popularity	9.47 (±3.94)	8.31 (±3.96)	8.90 (±2.89)	8.27 **
Appeareance	10.51 (±4.64)	8.87 (±4.90)	9.70 (±4.59)	12.77 ***
Values and Interests	7.26 (±2.97)	7.06 (±3.05)	7.17 (±3.02)	0.42

Note: ^a^ Number (and % in parentheses) for categorical data. ^b^ Means (and standard deviations in parentheses) for interval data. ** *p* < 0.01, *** *p* < 0.001.

**Table 3 behavsci-15-00434-t003:** Correlational analysis between variables.

	1.	2.	3.	4.	5.	6.	7.	8.	9.
1. Age									
2. Screen Time	0.10 *	1							
3. Trait EI	−0.05	−0.26 ***	1						
4. EDs risk	0.04	0.15 **	−0.39 ***	1					
5. PSS	0.00	−0.04	0.48 ***	−0.15 **	1				
6. Connection	0.05	0.21 ***	−0.29 ***	0.22 ***	−0.09	1			
7. Popularity	−0.03	0.24 ***	−0.18 ***	0.21 ***	−0.03	0.57 ***	1		
8. Appearance	−0.00	0.19 ***	−0.31 ***	0.27 ***	−0.15 **	0.61 ***	0.64 ***	1	
9. Interests	0.10 *	−0.02	0.00	0.13 **	0.04	0.22 ***	0.18 ***	0.19 ***	1

Note. Trait EI = global scores of Trait Emotional Intelligence; EDs risk = eating disorder risk; PSS = perceived social support; connection = motivation for social media use, i.e., connection with others; popularity = motivation for social media use; appearance = motivation for social media use; interests = motivation for social media use, i.e., values/interests. * *p* < 0.05. ** *p* < 0.01. *** *p* < 0.001.

**Table 4 behavsci-15-00434-t004:** Paths, effects, and confidence intervals of moderated–mediation models.

Model	Paths	R^2^	F	B	SE	z	95% CI
Model 1		0.18	21.35 ***				
	Trait EI → ED risk			−10.92 ***	3.05	−3.57	[−17.11; −5.14]
	Popularity → ED risk			0.37 **	0.12	3.06	[0.14; 0.61]
	PSS → ED risk			−0.35	0.19	−1.77	[−0.75; 0.01]
	Trait EI × PSS → ED risk			0.08 *	0.04	1.97	[0.01; 0.17]
	Trait EI → Popularity			−0.09 ***	0.25	−3.72	[−1.44; −0.44]
	Trait EI → Popularity → ED risk			−0.35 *	0.15	−2.36	[−0.65; −0.06]

Model 2		0.19	21.98 ***				
	Trait EI → ED risk			−11.03 ***	3.05	−3.61	[−17.02; −5.04]
	Appearance → ED risk			0.37 ***	0.11	3.37	[0.15; −0.58]
	PSS → ED risk			−0.36	0.19	−1.86	[−0.75; 0.02]
	Trait EI × PSS → ED risk			0.09 *	0.04	2.10	[0.00; 0.17]
	Trait EI → Appearance			−1.84 ***	0.28	−6.52	[−2.40; −1.30]
	Trait EI → Appearance → ED risk			−0.68 **	0.22	−3.00	[−1.13; −0.23]

Model 3		0.17	20.23 ***				
	Trait EI → ED risk			−11.22 ***	3.07	−3.65	[−17.42; −5.37]
	Connection → ED risk			0.40 *	0.17	2.34	[0.07; 0.75]
	PSS → ED risk			−0.37	0.19	−1.87	[0.77; 0.00]
	Trait EI × PSS → ED risk			0.09 *	0.04	2.08	[0.08; 0.18]
	Trait EI → Connection			−1.08 ***	0.18	−6.04	[−1.43; −0.73]
	Trait EI → Connection → ED risk			−0.44 *	0.20	−2.18	[−0.84; −0.05]

Model 4		0.18	21.61 ***				
	Trait EI → ED risk			−12.51 *	3.05	−4.09	[−18.49; −6.52]
	Values/interests → ED risk			0.05	0.15	3.19	[0.19; 0.82]
	PSS → ED risk			−0.42	0.19	−2.15	[−0.81; −0.03]
	Trait EI × PSS → ED risk			0.10 *	0.04	2.36	[0.02; 0.19]
	Trait EI → Values/interests			0.03	0.19	0.16	[−0.35; 0.41]
	Trait EI → Values/interests → ED risk			0.01	0.10	0.16	[−18; 0.21]

Note. Trait EI = global scores of Trait Emotional Intelligence; ED risk = eating disorder risk; PSS = perceived social support; connection = motivation for social media use, i.e., connection with others; popularity = motivation for social media use; appearance = motivation for social media use; interests = motivation for social media, i.e., values/interests. * *p* < 0.05. ** *p* < 0.01. *** *p* < 0.001.

## Data Availability

Data are available upon request due to privacy restrictions.

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
