# Peer review of "The Relationship Between Emotional Intelligence and the Risk of Eating Disorders Among Adolescents: The Mediating Role of Motivation for the Use of Social Media and Moderation of Perceived Social Support"

_behavsci, 2025, doi:10.3390/bs15040434_

Round 1

Reviewer 1 Report

Comments and Suggestions for Authors

The manuscript is an original article correlating emotional intelligence and eating disorders risk in adolescents with a focus on the moderating effect of social media motivation use and social support. The article is interesting and highly relevant. The authors analysed 388 adolescents in Italy for their EAT-26, MSPSS, TEIQue-ASF, and MSMU. This study shows that motivations for popularity and appearance mediate the EI levels and EAT-26 scores.

Abstract:

  • The abstract does not contain adequate information about the article. I recommend revising it by condensing the background to 2-3 sentences while incorporating more details regarding the methods and results, specifically the date and location of the study, as well as the strength and significance of the findings.

Background:

  • To make it clearer for the readers, I suggest including a table that compares the trait EI and ability EI, along with their relationship with EDs.
  • Please use a consistent abbreviation, either ED or EDs for eating disorders.
  • Each abbreviation should be mentioned only once in the first appearance (i.e. PSS, EI)
  • Source for sentence in line 66

Methods:

  • I suggest adhering to established reporting guidelines, such as STROBE, and including the checklist as a supplementary document.
  • Please include the inclusion and exclusion criteria.
  • Please explain how to determine the sample size in this study.
  • Please state how many schools are involved, and, if feasible, the characteristics of the schools (e.g. private/public, rural/urban areas)
  • Please specify the month & year during which participant recruitment took place.
  • Did the authors involve patients and public involvement in this research? Please provide further information.

Results:

  • Line 225: An unclear long sentence, consider dividing it into several sentences.
  • Do the authors have information on how many screen time hours participants have in low and high EAT-26 scores? I would suggest considering including the information, as this would be an interesting discussion point to be explored further, if possible.

Discussion:

  • Line 290: Although the study's findings show a correlation, this does not mean a causational relationship. The authors might consider rephrasing the sentence.
  • Do the authors have data on the proportion of participants diagnosed with EDs? If any participants were diagnosed with EDs before participating, the results and discussion should be interpreted with caution, as this could significantly impact the EAT-26 outcomes. Additionally, there was no clear explanation whether the participants had pre-existing EDs before their engagement with social media or if the social media use preceded the development of EDs. Please provide clarification regarding this concern.
  • Paragraph in line 335. I would suggest rewriting the paragraph, as it could lead to overclaim. As mentioned in the limitation, this study is a cross-sectional study that analyses a correlation.

Reviewer 2 Report

Comments and Suggestions for Authors

The paper entitled “ behavsci-3533566_The Relationship Between Emotional Intelligence and Risk of Eating Disorders among Adolescents: The Mediating Role of otivation for Use of Social Media and Moderation of Perceived Social Support“ is submitted to the Section “Developmental Psychology “, as part of the Special Issue “ Emotional Intelligence and Psychological Well-Being in Children and Adolescents “. This study aligns with the scope of this section.

The research addresses a highly relevant topic, aiming to advance our understanding of the theoretical relationships between trait Emotional Intelligence (EI), social media motivations, and perceived social support. Ultimately, it seeks to contribute to more targeted interventions aimed at reducing the risk of eating disorders among adolescents.

The abstract needs to be rewritten, as the available space should summarise the key content of the paper. Specifically, it should include the study’s precise objective, the methodology with its design, the results with quantitative data, and the conclusion.

The introduction is well-structured, drawing on relevant literature to justify the study. However, the research objective could be stated more explicitly.

In the Materials and Methods section, the study design needs to be specified. It would also be beneficial to indicate whether a sample size calculation was performed to ensure the study adequately addresses its objective. If such a calculation was conducted, it should be included.

In the Results section, Table 2 includes abbreviations that should be defined in a footnote to ensure the table is self-explanatory. The same applies to Table 3. Additionally, Table 3 shows considerable variability in the confidence interval, suggesting that the sample size should be increased. In the Discussion section, the sample size should be acknowledged as a limitation of the analysis.

The discussion is well-developed, offering a critical reflection on the findings in relation to existing literature. However, it is important to consider that the analysis used requires a larger sample size. Consequently, the results should be regarded as preliminary, and the limitations of this aspect should be further discussed.

Round 2

Reviewer 1 Report

Comments and Suggestions for Authors

The authors have addressed most of the suggestions, but one remaining suggestion has not been addressed. The STROBE checklist provided is not completed. Please include a column for the lines/pages where the information was provided in the study. The authors could refer to the "STROBE Checklist (fillable)" from www.strobe-statement.org.

Author Response

Thank you for giving us the opportunity to resubmit the manuscript. Below, you can find a point-by point answer to your comments.

Comment 1 The authors have addressed most of the suggestions, but one remaining suggestion has not been addressed. The STROBE checklist provided is not completed. Please include a column for the lines/pages where the information was provided in the study. The authors could refer to the "STROBE Checklist (fillable)" from www.strobe-statement.org.

Author Response:  We thank the reviewer for this final, important remark. In response, we have now fully completed the STROBE checklist using the official fillable version provided at www.strobe-statement.org. As requested, we have included a column indicating the specific sections and page numbers in our manuscript where each item is addressed. The updated checklist is now submitted as a supplementary document alongside the manuscript.